# Epitaxy, exfoliation, and strain-induced magnetism in rippled Heusler membranes

Dongxue Du[1], Sebastian Manzo[1], Chenyu Zhang [1], Vivek Saraswat[1], Konrad T. Genser[2], Karin M. Rabe[2], Paul M. Voyles [1], Michael S. Arnold [1] & Jason K. Kawasaki [1✉]

Single-crystalline membranes of functional materials enable the tuning of properties via extreme strain states; however, conventional routes for producing membranes require the use of sacrificial layers and chemical etchants, which can both damage the membrane and limit the ability to make them ultrathin. Here we demonstrate the epitaxial growth of the cubic Heusler compound GdPtSb on graphene-terminated $Al_2O_3$ substrates. Despite the presence of the graphene interlayer, the Heusler films have epitaxial registry to the underlying sapphire, as revealed by x-ray diffraction, reflection high energy electron diffraction, and transmission electron microscopy. The weak Van der Waals interactions of graphene enable mechanical exfoliation to yield free-standing GdPtSb membranes, which form ripples when transferred to a flexible polymer handle. Whereas unstrained GdPtSb is antiferromagnetic, measurements on rippled membranes show a spontaneous magnetic moment at room temperature, with a saturation magnetization of 5.2 bohr magneton per Gd. First-principles calculations show that the coupling to homogeneous strain is too small to induce ferromagnetism, suggesting a dominant role for strain gradients. Our membranes provide a novel platform for tuning the magnetic properties of intermetallic compounds via strain (piezomagnetism and magnetostriction) and strain gradients (flexomagnetism).

[1] Materials Science and Engineering, University of Wisconsin-Madison, Madison, WI, USA. [2] Department of Physics and Astronomy, Rutgers University, New Brunswick, NJ, USA. ✉email: jkawasaki@wisc.edu

Membranes are a powerful platform for flexible devices and for tuning properties via strain and strain gradients[1–7]. In contrast to the uniform strain of epitaxial films, strain in membranes can be applied dynamically, anisotropically, in gradient form, and at larger magnitudes. For example, recent experiments on ultrathin oxide membranes demonstrate the application of extreme uniaxial strain of 8%[1], whereas the maximum strain possible in an epitaxial thin film is typically no more than 3% before plastic deformation. In addition, membranes enable the application of strain gradients, which are difficult to control for a film that is rigidly clamped to a substrate (Fig. 1).

Magnetism is particularly attractive for tuning via strain in the membrane form. The coupling between magnetism and strain $\epsilon$, i.e., piezomagnetism ($M \propto \epsilon$) and magnetostriction ($M^2 \propto \epsilon$), is widely used to tune the magnetic properties of thin films[8–11]. In contrast, the magnetic coupling to strain gradients, i.e., flexomagnetism ($M \propto \nabla\epsilon$), has been theoretically predicted[12,13], but to our knowledge has not experimentally demonstrated. This is due, in part, to difficulties in synthesizing and controlling strain gradients in nanostructures and membranes. Conventional techniques for fabricating single-crystalline membranes require etching of a sacrificial layer[2,3,5,14], which requires a detailed knowledge of etch chemistry and limits the ability to make ultrathin membranes of air-sensitive materials.

We demonstrate the etch-free epitaxial synthesis and exfoliation of Heusler membranes and show that rippled membranes induce magnetic ordering, turning the antiferromagnetic half Heusler compound GdPtSb into a ferro- or ferrimagnet. Heusler compounds are a broad class of intermetallic compounds with tunable magnetic textures[15], topological states[16–18], and novel superconductivity[19,20]. In contrast with conventional membrane synthesis techniques, which require etching of a sacrificial buffer layer[1–4,14], our use of a monolayer graphene decoupling layer allows GdPtSb membranes to be mechanically exfoliated, bypassing the need for detailed knowledge of etch chemistries. Our approach is akin to "remote epitaxy," which has recently been demonstrated for the growth of compound semiconductor[21,22], transition metal oxide[7], halide perovskite[23], and elemental metal films[24]. Here, we show that similar approaches apply to intermetallic quantum materials. We find that the large strains and strain gradients in rippled GdPtSb membranes drive an antiferromagnet to ferri- or ferromagnet transition. First-principles calculations show that the coupling to homogeneous strain is too small to induce ferromagnetism,

suggesting a dominant role for strain gradients, which would make this system the first experimental example of flexomagnetic coupling. Our work opens a new platform for driving ferroic phase transitions via complex strained geometries.

## Results

**Epitaxy of GdPtSb on graphene-terminated sapphire.** Our concept relies on the weak van der Waals interactions of monolayer graphene to enable epitaxial growth and exfoliation of a membrane from a graphene-terminated single-crystalline substrate. Figure 2a shows a schematic heterostructure, which consists of cubic GdPtSb (space group F$\bar{4}$3m), polycrystalline monolayer graphene, and a single-crystalline Al$_2$O$_3$ substrate in (0001) orientation. GdPtSb crystallizes in the cubic half Heusler structure, the same structure as the antiferromagnetic Weyl semimetal GdPtBi[16,25,26]. Since the layer-transferred graphene

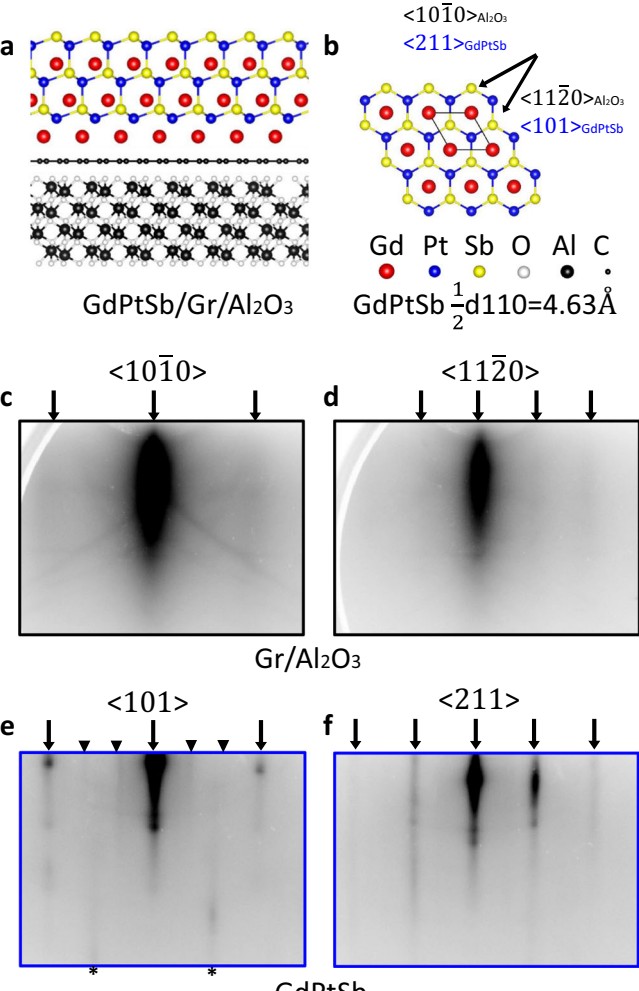

**Fig. 2 Epitaxy of GdPtSb on graphene-terminated Al$_2$O$_3$ (0001). a** Schematic crystal structure of (111)-oriented cubic GdPtSb on graphene/Al$_2$O$_3$ (0001). The lattice parameter for Al$_2$O$_3$ is $a = 4.785$ Å and that of GdPtSb is $\frac{1}{2}d_{110} = 4.53$ Å, corresponding to a mismatch of 2.7%. **b** In-plane crystal structure of GdPtSb (111). **c, d** Reflection high-energy electron diffraction (RHEED) patterns of graphene on Al$_2$O$_3$ after a 700 °C annealing. Black arrows mark the underlying Al$_2$O$_3$ reflections. **e, f** RHEED patterns for the GdPtSb film. Black arrows mark the bulk reflections. Triangles mark superstructure reflections from a 3 × surface reconstruction. Additional reflections (black asterisks) are observed that correspond to a second domain rotated by ±30°.

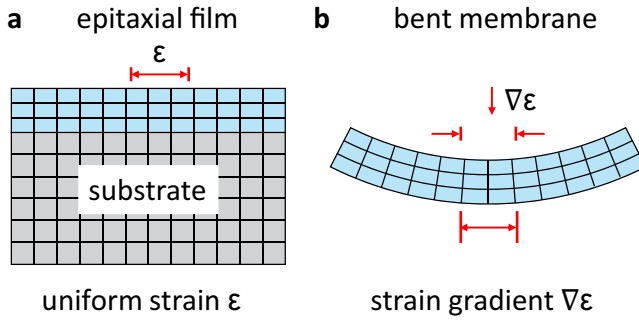

**Fig. 1 Modes of strain accessible to epitaxial films vs free-standing membranes. a** Piezomagnetism and magnetostriction are magnetic responses to uniform strain ($\epsilon$) and are accessible in thin films. **b** Flexomagnetism is the response to strain gradients ($\nabla\epsilon$) and is accessible in ultrathin membranes.

has randomly oriented polycrystalline domains, if the primary interactions are between GdPtSb film and graphene, then a polycrystalline film is expected. If, on the other hand, the primary interactions are between GdPtSb film and the underlying substrate, then an epitaxial film is expected. Given the recent demonstration of semi-lattice transparency of graphene during GaAs/graphene/GaAs "remote epitaxy"[21], we expect epitaxial registry between GdPtSb and the underlying sapphire to dominate. Single-crystalline GdPtSb membranes can then be exfoliated and strained in complex geometries.

To synthesize the GdPtSb/graphene/Al$_2$O$_3$ (0001) heterostructures, we first transfer polycrystalline monolayer graphene onto a pre-annealed Al$_2$O$_3$ (0001) substrate using standard wet transfer techniques ("Methods"). The graphene is grown on copper foil by chemical vapor deposition. Raman spectroscopy and atomic force microscopy indicate clean transfers with long-range coverage and minimal point defects or tears in the graphene (Supplementary Fig. S1). We then anneal the graphene/sapphire samples at 700 °C in ultrahigh vacuum ($P < 2 \times 10^{-10}$ Torr) to clean the surface. At this stage, the reflection high-energy electron diffraction (RHEED) pattern shows a bright but diffuse specular reflection compared to bare sapphire[27], which we attribute to diffuse scattering from the randomly oriented top graphene layer (Fig. 2c, d). There are weak diffraction streaks at the +1 and −1 positions (arrows), which we attribute to the underlying sapphire substrate.

GdPtSb films were grown by molecular beam epitaxy (MBE) on the graphene/Al$_2$O$_3$ at 600 °C, using conditions similar to ref.[27] ("Methods"). The streaky RHEED patterns for GdPtSb indicate growth with the epitaxial registry to the underlying sapphire substrate (Fig. 2e, f, black arrows). For beam oriented along $\langle 101 \rangle_{GdPtSb}$, we also observe superstructure reflections corresponding to a 3× surface reconstruction (Fig. 2e, triangles), indicating a well-ordered surface. In addition to the expected streaks for a hexagon-on-hexagon epitaxial relationship (black arrows), we observe faint secondary streaks marked by asterisks. The $\Delta Q$ spacing between these streaks differs from the main reflections (arrows) by a factor of $\sqrt{3}$, suggesting the presence of two domains that are rotated by 30°: one domain with the expected $\langle 101 \rangle_{GdPtSb} \parallel \langle 11\bar{2}0 \rangle_{Al_2O_3}$ epitaxial relationship, and the other rotated by ±30° around the Al$_2$O$_3$ [0001] axis with the epitaxial relationship $\langle 101 \rangle_{GdPtSb} \parallel \langle 10\bar{1}0 \rangle_{Al_2O_3}$.

XRD pole figure and cross-sectional TEM measurements confirm the presence of these two epitaxial domains. Figure 3c shows a pole figure of the GdPtSb 220 and Al$_2$O$_3$ 10$\bar{1}$4 reflections. We observe two sets of domains, one corresponding to the expected $\langle 101 \rangle_{GdPtSb} \parallel \langle 11\bar{2}0 \rangle_{Al_2O_3}$ relationship, and the other set rotated by ±30°. In contrast, GdPtSb films grown directly on sapphire do not show the ±30° domains (Supplementary Fig. S2). We speculate that the second domain forms for heterostructures with graphene because the weak interactions across the graphene change the balance between the energy of interfacial bonding and the strain energy, favoring small strains via a lattice rotation. A 0° GdPtSb domain has a lattice mismatch with a sapphire of 2.7%, whereas a ±30° rotation corresponds to a $(3, 3)_{GdPtSb} \parallel (5, 0)_{Al_2O_3}$ superstructure with a mismatch of only −1.5%. Here, we write the GdPtSb lattice vectors in hexagonal coordinates, where ($\mathbf{a}' \parallel 10\bar{1}, \mathbf{b}' \parallel 1\bar{1}0$). Rotational domains have also been observed for GaN films grown on monolayer h-BN/GaN (0001)[22] and for Cu films growth on monolayer graphene/Al$_2$O$_3$ (0001)[24]. However, in those cases, the presence of the second domain was attributed to an epitaxial relationship between the film (Cu or GaN) and the 2D monolayer (h-BN or graphene), while the primary domain results from an epitaxial relationship between the film and the substrate. In the present case of ±30° domains of GdPtSb, we rule out a long-range epitaxial relationship to the

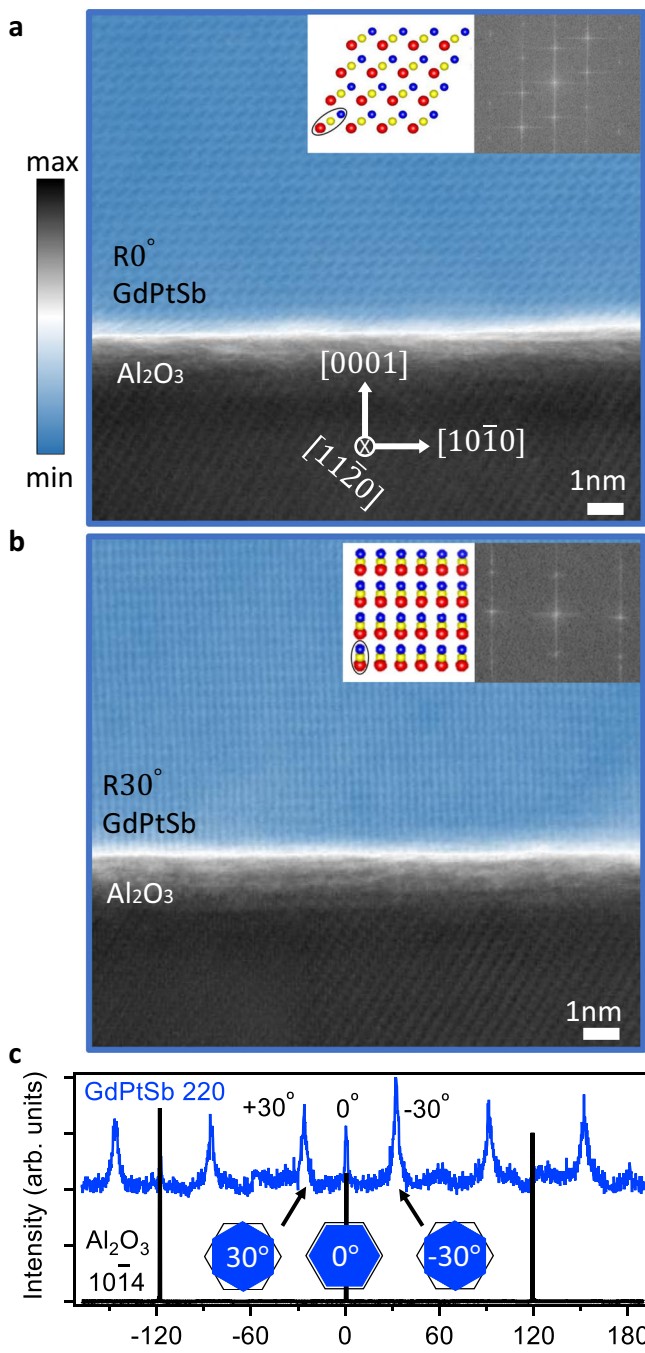

**Fig. 3 Heusler/graphene/sapphire interface. a**, **b** STEM image of the GdPtSb/graphene/Al$_2$O$_3$ interface for the two different GdPtSb domains, as viewed along the $\langle 11\bar{2}0 \rangle_{Al_2O_3}$ zone axis. We used log scale false-color image to simultaneously visualize the film and substrate for better contrast. The insets show the schematic crystal structures and fast Fourier transforms of the STEM images. **c** In-plane $\phi$ scan of GdPtSb 220 reflections and Al$_2$O$_3$ 10$\bar{1}$4 reflections for GdPtSb sample grown on graphene/Al$_2$O$_3$. $\Delta\phi = \pm 30°$ are the in-plane angles between the $\langle 101 \rangle$ direction of GdPtSb and the $\langle 11\bar{2}0 \rangle$ direction of Al$_2$O$_3$ substrate.

graphene because our graphene barrier is polycrystalline, and thus if there were an epitaxial relationship between GdPtSb and polycrystalline graphene, a large distribution of GdPtSb domain orientations would result.

Annular bright-field (ABF) scanning transmission electron microscopy (STEM) images in Fig. 3a, b confirm the existence of

these two sets of domains. In these images, the atomic structure is resolved a few nanometers away from the interface. It is difficult to resolve the registry at the GdPtSb/graphene/sapphire interface, which we attribute to partial film delamination during TEM sample preparation. The STEM image of the $\Delta\phi = 0$ domain consists of arrays of tilted spindle-shaped dark spots, each of which represents a combination of one Gd atomic column, one Pt atomic column, and one Sb atomic column. For the STEM image of the $\pm30°$ rotated GdPtSb, the spindle-shaped spots are aligned along the vertical direction corresponds to clusters of Gd-Pt-Sb atomic columns, as shown by the inset cartoon.

**Membrane exfoliation.** Heusler films grown on graphene/$Al_2O_3$ can be mechanically exfoliated to yield free-standing membranes, without the need for a metal stressor release layer. Figure 4a shows $\theta - 2\theta$ scans of GdPtSb, before and after exfoliation. Before exfoliation, we observe all of the expected 111, 222, 333, and 444 reflections. The rocking curve width of the 111 reflections before exfoliation is 15.4 arc second, indicating a high-quality film (Supplementary Fig. S3). After mechanical exfoliation (blue curve), we observe all of the expected GdPtSb reflections and none from the substrate, indicating a high crystallographic quality of the exfoliated membranes.

We find that the best exfoliation with minimal cracks is performed by adhering the sample film-side down to a rigid glass slide using Crystalbond and then prying off the substrate. The resulting membranes adhered to the glass slide have minimal long-range tears, as shown in Fig. 4b and Supplementary Fig. S5. It is also possible to exfoliate by adhering Kapton or thermal release tape directly to the film and peeling off the membrane; however, the bending during this peeling process produces microtears (Supplementary Fig. S4). An important aspect of this system is that unlike the compound semiconductor[21] or oxide[28] systems grown by "remote epitaxy," in this Heusler/graphene/$Al_2O_3$ system, no metal stressor layer was required in order to perform the exfoliation. As a control, GdPtSb films grown directly on sapphire could not be exfoliated by these methods.

**Strain-induced magnetism in rippled membranes.** We now demonstrate that the strains and/or large strain gradients in rippled Heusler membranes induce magnetic ordering, transforming antiferromagnetic GdPtSb films into ferro- (or ferri-) magnetic rippled membranes at room temperature. Figure 5a shows the magnetization $M$ versus applied magnetic field $H$ for a relaxed epitaxial GdPtSb film on $Al_2O_3$ (green) and for rippled membranes on two different polymer handles (blue and red). The unstrained film shows a weak linear $M(H)$ dependence. Temperature-dependent susceptibility measurements indicate that the films become antiferromagnetic at a Néel temperature of ~12 K (Supplementary Fig. S7). This behavior is consistent with the closely related GdPtBi, a $G$-type antiferromagnet ($T_N \sim 8.5$ K) in which local Gd moments are aligned ferromagnetically within (111) planes and the planes are aligned antiferromagnetically with neighboring planes[29]. Our first-principles calculations are consistent with an antiferromagnetic ground state for GdPtSb (Supplementary Fig. S9).

In contrast, rippled GdPtSb membranes on polyimide (blue) and polyurethane (red) layers show a spontaneous magnetization and hysteresis loops characteristic of ferrimagnetic (FiM) or ferromagnetic (FM) order. The saturation magnetization in the most rippled membrane on polyurethane is $5.2\,\mu_B$ per Gd atom, approaching the ~$7\mu_B$/Gd ferromagnetic limit (Fig. 5a). We observe a systematic dependence of the saturation magnetization (and magnetic susceptibility) on the ripple aspect ratio (height/width, Fig. 5c–f), suggesting that the origins of the magnetic order are

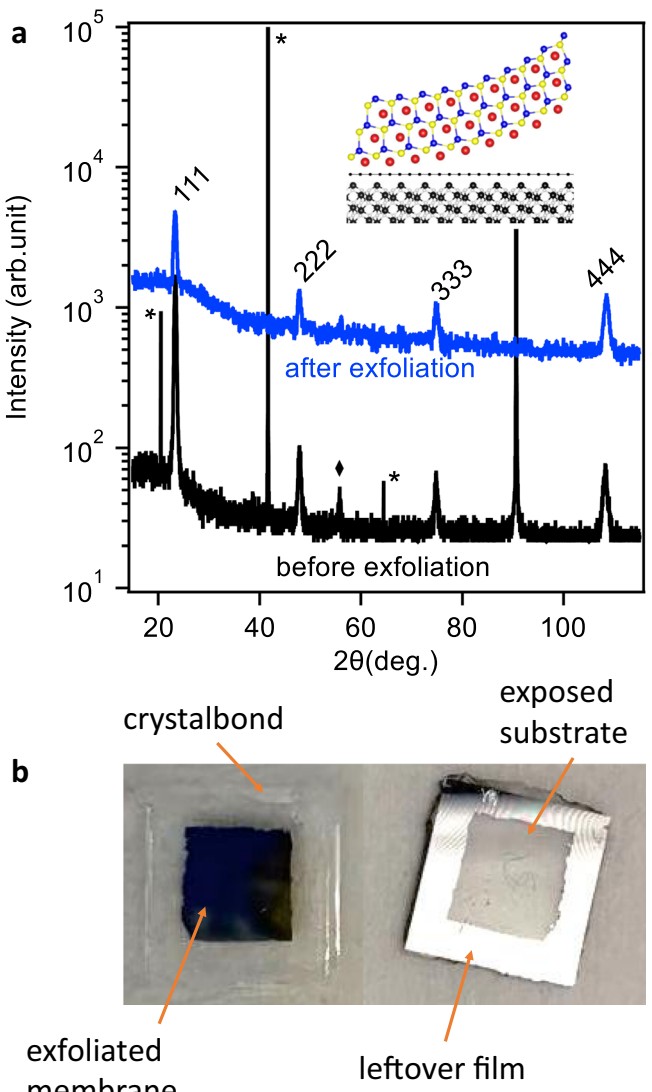

**Fig. 4 Crystallography and exfoliation of flat GdPtSb membranes. a** XRD $\theta$–$2\theta$ (Cu $K\alpha$) scans before exfoliation (black) and after exfoliation (blue). Sapphire substrate reflections are marked by asterisks. In addition to the epitaxial *lll*-type reflections, a small 004 reflection is observed and marked with a diamond. **b** Photos of GdPtSb membrane exfoliation from a $1 \times 1\,cm^2$ substrate. The left image shows a flat exfoliated membrane bonded to a glass slide. The right image shows the substrate side after exfoliation. In the region that had been covered with graphene (~$6 \times 6\,mm^2$ square at center), the GdPtSb membrane has been exfoliated to yield a transparent sapphire substrate. In the region not covered by graphene, the GdPtSb film remains adhered to the substrate.

ripple-induced strain or strain gradients, as opposed to extrinsic effects. These rippled membranes are made by adhering the membrane side of a flat membrane/crystalbond/glass slide stack to polymers with different thermal expansion coefficients (polyimide $\alpha = 34 \times 10^{-6}\,K^{-1}$ or polyurethane $\alpha = 57 \times 10^{-6}\,K^{-1}$). Melting the crystalbond on a hot plate releases the membrane/polymer bilayer. Ripples form upon release and cooling (Supplementary Fig. S6). We find that membranes on polymers with larger thermal expansion coefficients have larger ripple heights.

Our observation of ripple-induced magnetism in Heusler lies in contrast with previous studies of strain coupling in magnetic Heusler compounds. Previous studies have focused on the strain tuning of magnetic anisotropy, either in epitaxial films[30] or by

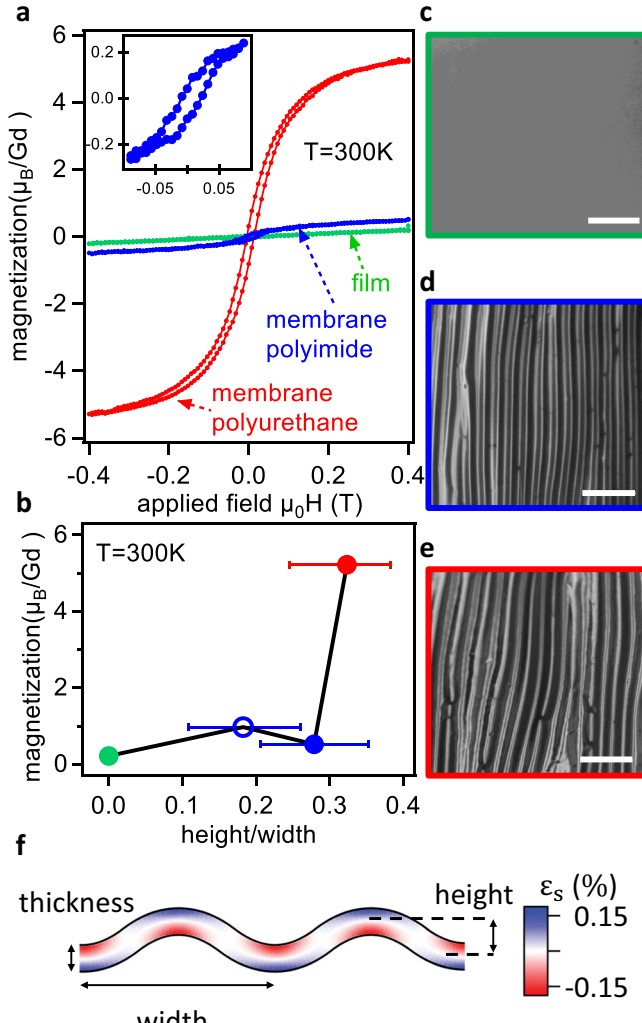

**Fig. 5 Strain-induced magnetism in rippled GdPtSb membranes. a** Magnetization of GdPtSb thin film directly grown on a sapphire substrate (green), exfoliated GdPtSb membrane adhered to polyimide (blue) and polyurethane (red). The applied field $H$ is oriented out-of-the sample plane, i.e., the global [111] direction. Both membranes have a thickness of 14 nm. **b** Room-temperature magnetization at 0.4 Tesla versus ripple aspect ratio (height/width). Relaxed film (green), rippled membrane on polyimide (filled and open blue circles), rippled membrane on polyurethane (red). The ripple height and width are measured by phase-contrast optical microscopy. The error bar represents the distribution of aspect ratio over several regions of the rippled membranes. **c–e** Optical microscope images of the film **c**, rippled membrane on polyimide **d**, and rippled membrane on polyurethane **e**, respectively. The scale bar is 100 μm. **f** Estimate of the strain $\epsilon_s$ for a sinusoidal membrane with peak to peak width (wavelength) 20 micron, ridge to valley height 5 micron, and thickness 14 nm, consistent with the dimensions of the membrane on polyurethane. Details in Supplementary Fig. S8.

bending in polycrystalline membranes on a flexible substrate[31]. In this study, we observe not a tunable anisotropy, but a strain-induced magnetic ordering. Other studies have investigated magnetoelastic coupling in ferromagnetic shape memory alloys[32,33], where the magnetic phase transition is coupled with a structural martensitic phase transition.

## Discussion

Due to the short-range nature of exchange interactions, subtle changes in bond length and local symmetry are expected to

modify the magnetic ground state, consistent with our observed AFM to FM (FiM) transition. For example, long-range and oscillatory Ruderman–Kittel–Kasuya–Yosida (RKKY) interactions were detected in some Mn-based Heuslers[34,35]. The RKKY interaction oscillates in sign and magnitude with the distance between magnetic atoms, exhibiting oscillation between ferromagnetic (FM) and antiferromagnetic (AFM) ordering[36–38]. However, the atomic-scale mechanism for ripple-induced magnetism in GdPtSb is not fully understood. A key question is whether the AFM to ferro- or ferrimagnetic ordering in our rippled membranes is driven by strain (piezomagnetism and magnetostriction) or strain gradients (flexomagnetism). Both strain and strain gradients are present in our rippled membranes, and both increase with the ripple aspect ratio. While piezomagnetism and magnetostriction have been widely studied, to our knowledge flexomagnetism has only been predicted[12,13], but not experimentally observed.

To estimate the relative contributions of strain and strain gradients, in Fig. 5f, we estimate the strain for a sinusoidal ripple in a membrane with the same thickness, height, and width as the rippled GdPtSb membrane on polyurethane. The magnitudes of strain $\epsilon_s$ along the sinusoidal paths are modest, with peak values of ±0.17%. In our first-principles calculations, ±0.17% is much too small a strain to induce a ferromagnetic or ferrimagnetic state, with a much larger strain of ~ 5% being required (Supplementary Fig. S9). In contrast, the estimated out-of-plane strain gradients $d\epsilon_s/dt$ are large, with peak values ±25% per micron (Supplementary Fig. S8). These large differences in magnitude suggest that the gradient term may dominate and that the observed behavior is flexomagnetic coupling; however, direct measurements of the strain state and direct calculations of the flexomagnetic response are required to fully understand the origins of magnetization. Our membranes provide a platform for the control and detailed understanding of the coupling between strain, strain gradients, and magnetism.

Our work also has strong implications for the remote epitaxial growth and other strain-induced properties in single-crystalline membranes. First, we expand the range of new functional materials that can be grown on graphene, to include intermetallic systems with mixed covalent and metallic bonding. Previous demonstrations of remote epitaxy have focused on transition metal oxides[28], halide perovskites[23], and compound semiconductors[21], which have more ionic bonding character. Second, strain and strain gradients in rippled membranes provide a platform for tuning other materials properties and ferroic orders, beyond magnetism. For example, it is an outstanding challenge to electrically switch the polarization of a polar metal due to strong charge screening. First-principals calculations suggest that strain gradients could be used to switch a polar metal via flexoelectric coupling[39], analogous to the possible flexomagnetism investigated here.

## Methods

**Synthesis of graphene**. Graphene was grown using thermal chemical vapor deposition of ultra-high purity $CH_4$ at 1050 °C on Cu foil. As-received copper foils (BeanTown Chemical number 145780, 99.8% purity) were cut into 1-inch-by-1-inch pieces and soaked in dilute nitric acid (5.7%) for 40 s followed by 3× DI water rinse followed by soaking in acetone and IPA to remove water from the surface. Dilute nitric acid helps remove the oxide and impurity particles from the surface. Foils were then dried under a gentle stream of air. Foils were subsequently loaded into a horizontal quartz tube furnace in which the furnace can slide over the length of the tube. Prior to monolayer graphene synthesis, the CVD chamber was evacuated to < $10^{-2}$ Torr using a scroll pump. The system was then back-filled with Ar and $H_2$, and a steady flow (331 sccm Ar, 9 sccm $H_2$) monitored by mass flow controllers was maintained at ambient pressure. The furnace was then slid to surround the samples, and annealed for 30 min. Then 0.3 sccm of P-5 gas (5% $CH_4$ in Ar) was flowed for 45 min so that a monolayer of graphene formed on the surface. To terminate the growth, the furnace was slid away from the samples, and

the portion of the quartz tube containing the samples was cooled to room temperature.

**Transfer of graphene to Al₂O₃**. Our graphene transfer procedure is a modified polymer-assisted wet transfer, similar to the transfer recipes studied in previous works[40]. (0001)-oriented Al₂O₃ substrates were prepared by annealing at 1400 °C for 10 h at atmospheric pressure in order to obtain a smooth, terrace-step morphology. To perform the graphene transfer, the graphene/Cu foils were cut into 6-mm-by-6-mm pieces and flattened using clean glass slides. Approximately 300 nm of 495K C2 PMMA (Chlorobenzene base solvent, 2% by wt., MicroChem) was spin-coated on the graphene/Cu foil substrate at 2000 RPM for 2 min and left to cure at room temperature for 24 h. Graphene on the backside of the Cu foil was removed via reactive ion etching using 90 W O₂ plasma at a pressure of 100 mTorr for 30 s. The Cu foil was then etched by placing the PMMA/graphene/Cu foil on the surface of an etch solution containing 1-part ammonium persulfate (APS-100, Transene) and 3-parts H₂O. After 10 h of etching at room temperature, the floating PMMA/graphene membrane was scooped up with a clean glass slide and sequentially transferred into three 30-min water baths to rinse the etch residuals. The PMMA/graphene membrane was then scooped out of the final water bath using the annealed sapphire substrate, to yield a PMMA/graphene/Al₂O₃ stack.

To remove water at the graphene/Al₂O₃ interface, samples were baked in the air at 50 °C for 5 min, then slowly ramped to 150 °C and baked for another 10 min. The PMMA is removed by submerging the sample in an acetone bath at 80 °C for 3 h. This is followed by an isopropanol and water rinse. The sample is indium bonded onto a molybdenum puck and outgassed at 150 °C for 1 h in a loadlock at a pressure $P < 5 \times 10^{-7}$ Torr before introduction to the MBE growth chamber. Finally, the graphene/Al₂O₃ sample is annealed at 400 °C for 1 h in the MBE chamber to desorb remaining organic residues and then annealed up to 700 °C immediately prior to the growth of GdPtSb.

**Molecular beam epitaxy of GdPtSb films**. GdPtSb films were grown on graphene/Al₂O₃ by molecular beam epitaxy (MBE) at a sample temperature of 600 °C, using conditions similar to ref. [27]. Gd flux was supplied by a thermal effusion cell. A mixture of Sb₂ and Sb₁ was supplied by a thermal cracker cell with a cracker zone operated at 1200 °C. The Pt flux was supplied by an electron beam evaporator. Fluxes were measured in situ using a quartz crystal microbalance (QCM) and calibrated with Rutherford Backscattering Spectroscopy. The Gd to Pt atomic flux ratio was maintained to be 1:1. Due to the high relative volatility of Sb, GdPtSb films were grown in an Sb adsorption-controlled regime with a 30% excess Sb flux. After the growth, films were capped in situ with 75 nm amorphous Ge to protect the surfaces from oxidation.

**Raman and atomic force microscopy of graphene/Al₂O₃**. Graphene transfer coverage and cleanliness is studied using field emission scanning electron microscopy (SEM) (Zeiss LEO 1530 Gemini). The graphene quality after the transfer is assessed via Raman spectroscopy using a 532 nm wavelength laser (Thermo Scientific DXR Raman Microscope). The laser power is kept below 5 mW in order to prevent damage to the graphene. The terrace-step morphology of the annealed sapphire substrates with and without graphene termination is analyzed by AFM (Bruker Multimode 8 SPM) in tapping mode.

**Scanning transmission electron microscopy**. The GdPtSb/graphene/Al₂O₃ sample was prepared for TEM using a Zeiss Auriga focused ion beam (FIB) with a final FIB polishing step with a 5 kV 100 pA Ga-ion beam. The sample surface was further polished for a higher smoothness in a Fishione 1040 Nanomill with a 900 eV Ar ion beam, to a thickness of ~80 nm. We did not seek to get the thinnest possible sample, as the film layer could peel off from the substrate when the sample is too thin. The TEM sample was kept under vacuum and cleaned in an Ibss GV10x DS Asher plasma cleaner for 10 min under 20 W to remove contaminations before being inserted into the TEM column.

A Thermo-Fisher Titan STEM equipped with a CEOS probe aberration corrector operated at 200 kV was used to collect STEM data. A 24.5 mrad semi convergence angle and an 18.9 pA current probe were used. A Gatan BF/ABF detector with 5.7 mrad and 22.8 mrad inner and outer collection angles was used to collect the annular bright-field (ABF) images.

**SQUID measurements**. Magnetic properties were measured using a Quantum Design MPMS SQUID (Superconducting Quantum Interference Device) Magnetometer. For the film data, we subtract a background measurement of the Al₂O₃ substrate. For the membrane samples, we subtract a background measurement of the polymer tape (polyimide or polyurethane) + Crystalbond.

**Density-functional theory calculations**. Density-functional theory (DFT) calculations were carried out with the ABINIT package. The pseudopotentials used were PAW JTH v1.1[41] within the local density approximation. Calculations were done on a conventional unit cell (simple cubic with four formula units per cell) with $6 \times 6 \times 6$ Monkhorst-Pak k-point mesh, a plane-wave cutoff of 25 Hartree,

with spin–orbit coupling included. The criterion for convergence was a potential residual of $<1.0 \times 10^{-8}$.

## Data availability
All data and code used in this paper will be made available upon reasonable request.

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

## Acknowledgements

Heusler epitaxial film growth and graphene transfers were supported by the Army Research Office (ARO Award number W911NF-17-1-0254) and the National Science Foundation (DMR-1752797). TEM experiments by C.Z. and P.M.V. were supported by the US Department of Energy, Basic Energy Sciences (DE-FG02-08ER46547), and used facilities are supported by the Wisconsin MRSEC (DMR-1720415). Graphene synthesis and characterization are supported by the U.S. Department of Energy, Office of Science, Basic Energy Sciences, under award no. DE-SC0016007. DFT calculations by K.T.G. and K.M.S. were supported by the Office of Naval Research award number ONR N00014-19-1-2073. We gratefully acknowledge the use of X-ray diffraction facilities supported by the NSF through the University of Wisconsin Materials Research Science and Engineering Center under Grant No. DMR-1720415. Writing and analysis were partially supported by the Air Force Office of Scientific Research (No. FA9550-21-0127).

## Author contributions

D.D. and J.K.K. conceived the project. D.D. performed the Heusler film growth, diffraction, and magnetic measurements. S.M. performed the graphene transfer and characterization. V.S. and M.S.A. synthesized the graphene. C.Z. and P.M.V. performed TEM measurements. K.T.G. and K.M.R. performed DFT calculations. J.K.K. supervised the project. D.D. and J.K.K. wrote the paper with feedback from all authors.

## Competing interests

The authors declare no competing interests.
