## [Peer Review File · Nature Communications]

REVIEWER COMMENTS

Reviewer #1 (Remarks to the Author):

The authors reported an interesting result regarding growing Heusler films on graphene/sapphire. The results are interesting and it will make a high impact in the community. However, a reviewer found out that interpretation of their result was somehow not clear and thus. I believe the manuscript will be completely suitable for Nature Communications after revising the issues listed below.

1. Remote epitaxy of metals on graphene-coated sapphire substrates has been already demonstrated by Lu's group [Z. Lu et al Nanotechnology 29, 44, (2018)] where they experimentally and theoretically show that Cu is remotely aligned to sapphire substrates through graphene. Later, W. Kong et al. [Nature Materials 17, 999–1004 (2018)], demonstrated that polarity influences epitaxial registry and ionically bonded materials penetrates more field through graphene. Those prior works already imply that metal alloys could read the patterns of ionically bonded Al₂O₃ substrates. In order to claim the possibility of lateral overgrowths from pinholes, direct evidences need to be provided. But this is not the case in the current form of manuscript. Authors may verify the case further if they can further perform DFT or KMC simulation to see if Heusler alloys can follow the potential fluctuation from sapphire substrates if possible.

2. The authors pointed out that direct epitaxy might happen at pinhole defects of the graphene. However, in CVD grown graphene, it is almost impossible for the pinholes to be digitally and uniformly distributed to offer artificial nucleation sites to finish up lateral growths to have perfect single-crystalline films. In addition, the success of exfoliation of the Heusler film without stressors also tells that majority was not by direct epitaxy as it does not make sense that strong covalent bonding through the pinholes can be broken without any additional energy. As this is very important message, authors must describe in detail about the exfoliation and transfer process of their Heusler films. Also please describe more about the yield of exfoliation and transfer. Also along this line, authors may be able to verify the distance between pinholes and distribution of pinholes in graphene indirectly by characterizing spalling marks on the exfoliated side (spalling marks appear after peeling if there is any direct epitaxy on the substrate through pinholes, see the extended figures at author's citation # [25]).

3. The authors discovered that in-plane twist was observed between RPtSb films and sapphire substrates. They need to clearly explain the reasons for this rotation. Refer the followings would be helpful: Z. Lu et al., Nanotechnology 29, 44, (2018) and Bae et al., Nature Nanotechnology, 15, 272 (2020).

4. Finally, it would be great if authors can demonstrate how useful their freestanding Heusler membranes are, such as killer applications of the freestanding Heuslers.

Reviewer #2 (Remarks to the Author):

Dear authors of the paper:

After carefully reading your paper, these are my suggestions:

The scientific topic of the paper "Epitaxy and exfoliation of free-standing Heusler membranes using a graphene interlayer" is of high interest and the results obtained are completed and satisfactory.

Here are some of the comments (related to the typos) after reading the work:

--- chemistry instead of chemsitry in line 26.

--- In FIG. S-3, the figure caption "Pole figure scans of (a) LaPtSb and (b) GdPtSb grown directly on Al₂O₃ without the graphene inter-layer." is not correlated to the graphs (a is b, and vice versa).

Reviewer #3 (Remarks to the Author):

In this work, the authors grow LaPtSb and GdPtSb films on graphene-covered Al₂O₃, then delaminate the films using thermal release tape to produce free-standing membranes. Using a combination of diffraction, microscopy, and transport measurements, the authors characterize the structure and properties of the thin films.

Overall, I thought this paper was well-written and overall technically well-justified, but I had difficulty seeing why it should be of general interest to a Nature Communications audience. The authors take an established method (remote epitaxy through graphene) and extend it to a new set of systems, Heusler compounds grown on sapphire. By itself, that represents nice work and is certainly worth publishing somewhere. But, the authors argue that the main innovations here are that the a) material has some rotational disorder that might enable moiré heterostructures from 3D materials, and that b) removal of the material with thermal release tape opens up the possibility of free-standing Heusler membranes. While both are technically true, the authors do not demonstrate measurements or theoretical predictions that show that either would represent an important advance. For example, what new device or capability could result from a free-standing Heusler membrane? I am skeptical in particular of the claim that moiré heterostructures of 3D materials are particularly interesting. Moiré structures are important in 2D materials because their properties are interface dominated, and as a result, it is possible to alter the bandstructure and other properties of 2D materials through control over the interface alignment; it is much less clear that this should be true in 30-40 nm thin films. More broadly, what is the difference between a moiré 3D heterostructure and a simple bicrystal separated by a grain boundary? I think for this paper to be of broad general interest, this work would need to be extended to demonstrate a new property or capability of the Heusler thin films. As it currently stands, this work would be a better fit for a more specialized journal, for example 2D materials or Advanced Functional Materials.

Dear Editor,

We thank the reviewers for their expert feedback. In the revised manuscript, we report strain- and strain gradient-induced magnetic ordering in rippled GdPtSb membranes, turning GdPtSb from an antiferromagnet to a ferri- or ferromagnet. These rippled membranes provide a novel platform for tuning ferroic order in quantum material membranes, via strain states that are not accessible in epitaxially clamped films or bulk crystals.

We believe this discovery satisfies the primary suggestions of Reviewers 1 and 3, to include a novel property of membranes. We also include details and morphological characterization of the exfoliated membranes, as suggested by Reviewer 1. Please see below for specific responses.

Reviewer #1 (Remarks to the Author):

The authors reported an interesting result regarding growing Heusler films on graphene/sapphire. The results are interesting and it will make a high impact in the community. However, a reviewer found out that interpretation of their result was somehow not clear and thus. I believe the manuscript will be completely suitable for Nature Communications after revising the issues listed below.

1. Remote epitaxy of metals on graphene-coated sapphire substrates has been already demonstrated by Lu's group [Z. Lu et al Nanotechnology 29, 44, (2018)] where they experimentally and theoretically show that Cu is remotely aligned to sapphire substrates through graphene. Later, W. Kong et al. [Nature Materials 17, 999–1004 (2018)], demonstrated that polarity influences epitaxial registry and ionically bonded materials penetrate more fields through graphene. Those prior works already imply that metal alloys could read the patterns of ionically bonded Al₂O₃ substrates. In order to claim the possibility of lateral overgrowths from pinholes, direct evidence needs to be provided. But this is not the case in the current form of manuscript. Authors may verify the case further if they can further perform DFT or KMC simulation to see if Heusler alloys can follow the potential fluctuation from sapphire substrates if possible.

We thank the reviewer for this outstanding point. In the third paragraph of the introduction we include the suggested citations by Lu et. al. and by Kong et. al.

We have decided to refocus this manuscript on the first Heusler membrane synthesis on graphene and the observation of strain-induced magnetism in rippled membranes. There are indeed many open questions regarding the atomic scale growth mechanisms of remote epitaxy, the role of defects, and what length scales are most relevant. Given the novelty of the synthesis in this new material system and the observation of strain-induced magnetism, we feel that the discussion of growth mechanisms belongs elsewhere, supported by a more detailed study at atomic length scales.

2. The authors pointed out that direct epitaxy might happen at pinhole defects of the graphene. However, in CVD grown graphene, it is almost impossible for the pinholes to be digitally and uniformly distributed to offer artificial nucleation sites to finish up lateral growths to have perfect single-crystalline films. In addition, the success of exfoliation of the Heusler film without stressors also tells that majority was not by direct epitaxy as it does not make sense that strong covalent bonding through the pinholes can be broken

without any additional energy. As this is very important message, authors must describe in detail about the exfoliation and transfer process of their Heusler films. Also please describe more about the yield of exfoliation and transfer. Also along this line, authors may be able to verify the distance between pinholes and distribution of pinholes in graphene indirectly by characterizing spalling marks on the exfoliated side (spalling marks appear after peeling if there is any direct epitaxy on the substrate through pinholes, see the extended figures at author's citation # [25]).

A detailed analysis of the exfoliated membranes, including small spalling marks, is now included in the main text Fig. 4b and in Supplemental Figure S-5 and S-6.

3. The authors discovered that in-plane twist was observed between RPtSb films and sapphire substrates. They need to clearly explain the reasons for this rotation. Refer the followings would be helpful: Z. Lu et al., Nanotechnology 29, 44, (2018) and Bae et al., Nature Nanotechnology, 15, 272 (2020).

Thank you for these references. Given the new focus on magnetism in GdPtSb, the small angle twist of LaPtSb is now beyond the scope of the current manuscript.

Regarding the 0 and 30 degree domains of GdPtSb, we add the following discussion on line 142: *“Rotational domains have also been observed for GaN films grown on monolayer h-BN/GaN (0001) [22] and for Cu films growth on monolayer graphene/Al₂O₃ (0001) [24]. However, in those cases the presence of the second domain was attributed to an epitaxial relationship between the film (Cu or GaN) and the 2D monolayer (h-BN or graphene), in addition to the primary domain that results from an epitaxial relationship between the film and the substrate. In the present case of +- 30 degree domains of GdPtSb, we rule out a long-range epitaxial relationship to the graphene because our graphene barrier is polycrystalline, and thus if there were an epitaxial relationship between GdPtSb and polycrystalline graphene, a large distribution of GdPtSb domain orientations would result.”*

4. Finally, it would be great if authors can demonstrate how useful their freestanding Heusler membranes are, such as killer applications of the freestanding Heuslers.

Thank you for this suggestion, which we feel has significantly broadened the impact of this work. In Fig. 5 we report the strain-induced magnetism of rippled GdPtSb membranes. We also discuss how the ripples in membranes can be used to tune other ferroic orders, such as ferroelectricity or the polarization of a polar metal.

Reviewer #2 (Remarks to the Author):

Dear authors of the paper:

After carefully reading your paper, these are my suggestions:

The scientific topic of the paper “Epitaxy and exfoliation of free-standing Heusler membranes using a graphene interlayer” is of high interest and the results obtained are completed and satisfactory.

Here are some of the comments (related to the typos) after reading the work:

--- chemistry instead of chemsity in line 26.

--- In FIG. S-3, the figure caption “Pole figure scans of (a) LaPtSb and (b) GdPtSb grown directly on Al₂O₃ without the graphene inter-layer.” is not correlated to the graphs (a is b, and vice versa).

Thank you. We have made the suggested typographical changes.

Reviewer #3 (Remarks to the Author):

In this work, the authors grow LaPtSb and GdPtSb films on graphene-covered Al₂O₃, then delaminate the films using thermal release tape to produce free-standing membranes. Using a combination of diffraction, microscopy, and transport measurements, the authors characterize the structure and properties of the thin films.

Overall, I thought this paper was well-written and overall technically well-justified, but I had difficulty seeing why it should be of general interest to a Nature Communications audience. The authors take an established method (remote epitaxy through graphene) and extend it to a new set of systems, Heusler compounds grown on sapphire. By itself, that represents nice work and is certainly worth publishing somewhere. But, the authors argue that the main innovations here are that the a) material has some rotational disorder that might enable moiré heterostructures from 3D materials, and that b) removal of the material with thermal release tape opens up the possibility of free-standing Heusler membranes. While both are technically true, the authors do not demonstrate measurements or theoretical predictions that show that either would represent an important advance. For example, what new device or capability could result from a free-standing Heusler membrane?

In Fig. 5 we report the strain-induced magnetism of rippled GdPtSb membranes. We also discuss how the ripples in membranes can be used to tune other ferroic orders, such as ferroelectricity or the polarization of a polar metal.

I am skeptical in particular of the claim that moiré heterostructures of 3D materials are particularly interesting. Moiré structures are important in 2D materials because their properties are interface dominated, and as a result, it is possible to alter the bandstructure and other properties of 2D materials through control over the interface alignment; it is much less clear that this should be true in 30-40 nm thin films. More broadly, what is the difference between a moiré 3D heterostructure and a simple bicrystal separated by a grain boundary? I think for this paper to be of broad general interest, this work would need to be extended to demonstrate a new property or capability of the Heusler thin films. As it currently stands, this work would be a better fit for a more specialized journal, for example 2D materials or Advanced Functional Materials.

The moiré pattern that we were originally envisioning refers to the small angle twist between LaPtSb and the underlying sapphire substrate (or graphene), which forms a planar interface. We were not thinking of a grain boundary between two adjacent LaPtSb domains. However, since we refocus the manuscript on magnetism in rippled membranes, the moiré discussion is now beyond the scope of this manuscript.

REVIEWERS' COMMENTS

Reviewer #1 (Remarks to the Author):

The authors did a great job on clarifying all concerns raised previously. The manuscript is read to be published in Nature Comm.

Reviewer #3 (Remarks to the Author):

The addition of the additional studies showing strain-induced changes in magnetism in the Heusler films addresses my main comments from the previous version. Overall, the revised version is significantly improved, and I am happy to recommend the paper for publication in Nature Communications.

Please see below for our point by point responses to the reviewers.

Reviewer #1 (Remarks to the Author):

The authors did a great job on clarifying all concerns raised previously. The manuscript is read to be published in Nature Comm.

We thank Reviewer 1 for their feedback.

Reviewer #3 (Remarks to the Author):

The addition of the additional studies showing strain-induced changes in magnetism in the Heusler films addresses my main comments from the previous version. Overall, the revised version is significantly improved, and I am happy to recommend the paper for publication in Nature Communications.

We thank Reviewer 3 for their feedback.